# Quaternized Diaminobutane/Poly(vinyl alcohol) Cross-Linked Membranes for Acid Recovery via Diffusion Dialysis

**DOI:** 10.3390/membranes11100786

**Published:** 2021-10-14

**Authors:** Muhammad Adnan Ashraf, Atif Islam, Muhammad Arif Butt, Hafiz Abdul Mannan, Rafi Ullah Khan, Kashif Kamran, Shahid Bashir, Javed Iqbal, Ahmed A. Al-Ghamdi, Abdullah G. Al-Sehemi

**Affiliations:** 1Institute of Polymer and Textile Engineering, University of the Punjab, Quaid-e-Azam Campus, Lahore 54590, Pakistan; adnanashrafm1@gmail.com (M.A.A.); hmannan.ipte@pu.edu.pk (H.A.M.); rkhan.icet@pu.edu.pk (R.U.K.); 2Institute of Chemical Engineering and Technology, University of the Punjab, Quaid-e-Azam Campus, Lahore 54590, Pakistan; drmarifbutt@yahoo.com; 3Department of Physics, University of Agriculture, Faisalabad 38040, Pakistan; k.kamran@uaf.edu.pk; 4Centre for Ionics University of Malaya, Department of Physics, Faculty of Science, University of Malaya, Kuala Lumpur 50603, Malaysia; 5Center of Nanotechnology, King Abdulaziz University, Jeddah 21589, Saudi Arabia; iqbaljavedch@gmail.com; 6Department of Physics, Faculty of Science, King Abdulaziz University, Jeddah 21589, Saudi Arabia; AGAMDI@kau.edu.sa; 7Department of Chemistry, Faculty of Science, Research Center for Advanced Materials Science (RCAMS), King Khalid University, Abha 61413, Saudi Arabia; agsehemi@kku.edu.sa

**Keywords:** diffusion dialysis, PVA, TEOS cross-linked, acid recovery, anion exchange membrane

## Abstract

Diffusion dialysis (DD) using anion exchange membranes (AEM) is an effective process for acid recovery and requires the preparation of suitable materials for AEMs, characterized by unique ions transport properties. In this work, novel AEMs composed of quaternized diaminobutane (QDAB) and poly(vinyl alcohol) (PVA) were cross-linked by tetraethoxysilane (TEOS) via the sol–gel process. The prepared AEMs were systematically characterized by Fourier-transform infrared (FTIR) spectroscopy, ion-exchange capacity (IEC) analysis, thermo gravimetric analysis (TGA), water uptake, linear expansion ratio (LER), and mechanical strength determination, scanning electron microscopy (SEM), and DD performance analysis for acid recovery using a hydrochloric acid/iron chloride (HCl/FeCl_2_) aqueous mixture and varying the QDAB content. The prepared AEMs exhibited IEC values between 0.86 and 1.46 mmol/g, water uptake values within 71.3 and 47.8%, moderate thermal stability, tensile strength values in the range of 26.1 to 41.7 MPa, and elongation from 68.2 to 204.6%. The dialysis coefficient values were between 0.0186 and 0.0295 m/h, whereas the separation factors range was 24.7–44.1 at 25 °C. The prepared membranes have great potential for acid recovery via diffusion dialysis.

## 1. Introduction

Wastewater is produced in various contexts including hydrometallurgy, metal refining, steel processing, aluminum etching, electroplating, and resin regeneration. It contains inorganic acids and metal ions [1,2]. The direct disposal of these acid effluents in the environment without treatment not only wastes useful acids but also is a great threat to human health and the environment [3,4,5]. The traditional methods that have been applied to treat acidic waste comprise extraction [6], ion exchange [7], electrodialysis [8], alkali neutralization [9], and pyrohydrolysis [10]. In comparison with all separation-based methods, anion exchange membrane (AEM)-based diffusion dialysis (DD) is highly preferable to treat acid wastes due to its energy efficiency, as electric power is only required for pumping the liquid streams into the dialyzer system and it is of easy installation, that does not require complex equipment, environment-friendly, because of no extra chemical agents are required, and cost-effective [3,11].

In the DD process, due to the difference in concentration gradient and the positive charge of the AEM itself, the anions pass through the AEM preferentially from the side with higher concentration to that with lower concentration [12,13]. The transport of counter-ions through the membrane is facilitated by the attractive force from fixed group in the membrane. Meanwhile, due to electrical neutrality, some co-ions with small hydration radius and low charge can always have high mobility and cross the membrane, which makes the DD process successful [14,15]. Till now, DD has been widely used in the recovery of inorganic acids including hydrochloric acid (HCl) [16], sulfuric acid (H_2_SO_4_) [17], nitric acid (HNO_3_) [18], and mixed inorganic acids [19].

The AEMs used to recover acid from wastes through diffusion dialysis contain predominantly cationic groups such as –NH_3_^+^, –NRH_2_^+^, –NR_2_H^+^, and –NR_3_^+^ [20,21,22,23]. Several anion exchange membranes have been prepared for acid recovery from various polymeric materials. Polymer-containing hydrophobic matrices include poly(vinyl chloride), poly(2,6-dimethyl-1,4-phenyleneoxide), poly(vinylidene fluoride), poly(sulfone), poly(styrene) [24,25,26], and hydrophilic PVA [27,28]. The major critical issue with hydrophobic membranes is their hydrophobicity, which restricts proton permeability during the DD process [29]. PVA has high hydrophilicity, is chemically and thermally stable, has excellent mechanical strength, excellent film forming ability, and a relatively low cost [30,31]. Along with these advantages, the presence of −OH groups with high hydrophilicity can enhance the mobility of hydrogen ions through hydrogen bonds in diffusion dialysis [32]. The available large amount of −OH groups in PVA, which allow for extraordinary reactions with small alkoxy silanes like tetraethoxysilane (TEOS) through the sol–gel process, enhances the mechanical and thermal properties, resulting in a dense, homogeneous, and compact membrane structure [33]. In order for a PVA matrix to act as an anion exchanger, cationic groups must be introduced into the membrane structure. The introduction of an anion exchange precursor into the structure followed by cross-linking is an efficient method to enhance the anion exchange capacity of membranes [34].

In many previously published works, various PVA-based AEMs membranes were synthesized for acid recovery, evaluating membranes performance via DD [7,35,36]. Emmanuel et al. published a series of doubly quaternized siloxane-cross-linked PVA hybrid AEMs prepared through the sol–gel technique for the recovery of acids by DD. The resulting membranes demonstrated high values of performance parameters such as dialysis coefficient, which varied between 0.030 and 0.0449 m/h, and separation factor of 20.9 to 32.3 at 25 °C [37]. Congliang et al. prepared glycidyltrimethylammonium chloride-cross-linked PVA-based membranes and investigated the effect of glycidyltrimethylammonium chloride on the recovery of the acid through DD. The dialysis coefficient value was in the range of 0.011–0.018 m/h, and the separation factor increased up to 21 [38]. Yonghui et al. studied the performance of PVA–silica hybrid anion exchange membranes and found that the separation factor was in the range of 15.9–21, and the acid recovery values were higher than those of the commercially available DF-120 membrane [32]. In similar studies, PVA-based AEMs showed better acid recovery performance and separation factor compared to the DF-120 membrane [23,39]. Hence, PVA-based hybrid membranes have been developed with different cross-linking agents and anion exchange groups to enhance their DD performance for acid recovery.

In this work, anion exchange hybrid membranes based on PVA and quaternized diaminobutane (QDAB) cross-linked with TEOS were synthesized through the sol–gel process and evaluated for DD performance in acid recovery. Glycidyltrimethylammonium chloride (GTMAC) has been widely used as an anion exchange material for acid recovery membranes. The compound 1,4-diaminobutane (DAB) was used as a junction between GTMAC and an anion exchange precursor (QDAB) mixed with PVA along with the TEOS cross-linker, then AEMs were fabricated. The innovation of this work is the dispersion of QDAB in PVA cross-linked membranes, which, according to the best of our knowledge, has never been reported for acid recovery via DD. QDAB possesses two ionic sites serving as fast charge carriers and can enhance the ionic diffusivity during DD. The cross-linking agent TEOS ensured the thermal and mechanical stability of the membrane by changing its morphological properties and hydrophilicity [40]. Moreover, Fourier-transform infrared spectroscopy (FTIR), IEC, water uptake, and swelling degree determination, thermo gravimetric analysis (TGA), scanning electron microscopy (SEM), and mechanical strength analysis of the prepared membranes were carried out. The membranes were finally tested for acid recovery and separation performance by conducting diffusion dialysis experiments using an HCl/FeCl_2_ aqueous mixture.

## 2. Experimental Details

### 2.1. Materials

PVA with average molecular weight of 72,000 g/mol and degree of hydrolyzation ≥ 98%, ferrous chloride (FeCl_2_·4H_2_O), sodium chloride (NaCl), hydrochloric acid (HCl), potassium permanganate (KMnO_4_) and sodium carbonate (Na_2_CO_3_) were purchased from Merck, Germany. The compounds 1,4-diaminobutane (DAB, 99%), glycidyltrimethylammonium chloride (GTMAC, ≥90%), and dimethyl sulfoxide (DMSO, 99%) were obtained from Sigma Aldrich, St. Louis, MO, USA. Tetraethoxysilane (TEOS, 98.5%) was purchased from Daejung chemical and metals Co. Ltd., Shiheung, Gyeonggi-do, South Korea. Other chemicals were of analytical grade and used without any additional purification.

### 2.2. Synthesis of Quaternized Diaminobutane (QDAB)

QDAB was prepared by the reaction of 1,4-diaminobutane and GTMAC in DMSO, as shown in reaction Figure 1. Firstly, 1 g of 1,4-diaminobutane (0.01134 mol) and 12 mL of DMSO as solvent were mixed in a 50 mL round-bottom flask with continuous magnetic stirring at 85 °C for 30 min. In this transparent reaction mixture, a calculated amount of GTMAC (3.44 g, 0.02269 mol) was added. Stirring was continued at 85 °C for 12 h to complete the reaction. A viscous and homogeneous solution of QDAB was obtained, cooled down to room temperature, and kept in a sealed bottle for further use.

### 2.3. Preparation of QDAB/PVA Membranes

The QDAB/PVA AEMs were prepared using the standard sol–gel process, as shown in the proposed reaction Figure 2. First, 1 g of PVA was dissolved in 19 g DMSO with continuous stirring for 4 h at 85 °C. A clear homogeneous solution of PVA (5 wt. %) was obtained and cooled down to 60 °C. A suitable amount of QDAB with respect to PVA in DMSO solution were separately stirred for 30 min at ambient conditions, and the QDAB solution was added to the prepared PVA solution. Then, 0.2 g of TEOS (20 wt. % to PVA) cross-linker along with a small quantity of 0.1 M HCl solution (as a catalyst) was carefully added in the reaction mixture. The reaction mixture was stirred at 60 °C for 24 h on a hot plate to complete the sol–gel reaction by the hydrolysis of silane. The resulting gel mixture was cast on a Petri dish and dried in a preheated oven at 60 °C for 24 h. The obtained membrane was carefully peeled off the Petri dish with the help of a knife blade. Finally for thermal cross-linking by heat treatment, the AEMs were heated from 70 °C to 130 °C at the heating rate of 10 °C/h and kept at 130 °C for 4 h to ensure a complete conversion into hybrid membranes. The thickness of the membrane was maintained around 165 to 190 μm. Five different membranes, mentioned in Table 1, were prepared by varying the amount of QDAB from 30 wt. % to 70 wt. % of PVA and were coded as QDAB-30, QDAB-40, QDAB-50, QDAB-60, and QDAB-70, respectively.

### 2.4. Membrane Characterizations

#### 2.4.1. Fourier-Transform Infrared (FTIR) Spectroscopy

FTIR spectroscopic analysis of the PVA hybrid membrane was evaluated with an IR Shimadzu, Prestige-21 (Japan) spectrometer attenuated with a Horizontal Attenuated Total Reflectance (HATR) equipment. The spectral range of the experiment was 4000–400 cm^−1^, and 100 scans were collected per spectrum at a resolution of 4 cm^−1^. The experiment was performed with air as its background.

#### 2.4.2. Ion-Exchange Capacity (IEC)

The IEC of the prepared membranes was determined to investigate the content of interchangeable ionic groups in the membranes using the Mohr method. A dried membrane sample was weighed and soaked in a 1.0 M NaCl solution for 24 h at room temperature to convert the ion-exchange groups into Cl^−^. After this, the sample was washed with deionized water to remove excessive NaCl and immersed in 50 mL of a 0.05 mol/L Na_2_SO_4_ aqueous solution for 48 h to replace the Cl^−^ ions by SO_4_^2−^ ions. The concentration of the released Cl^−^ ions in solution was determined by titration using a 0.05 mol/L AgNO_3_ solution, and K_2_CrO_4_ was used as an indicator [41]. The IEC (mmol/g) of the membrane was determined with Equation (1):(1)$$\mathrm{IEC}=\frac{C_{{\mathrm{AgNO}}_{3}}{\;V}_{{\mathrm{AgNO}}_{3}}}{W_{\mathrm{dry}}}$$
where, $C_{{\mathrm{AgNO}}_{3}}$, $V_{{\mathrm{AgNO}}_{3}}$, and W_dry_ represent the molar concentration (mol/L) of the AgNO_3_ solution, the volume (mL) of the AgNO_3_ solution consumed, and the dry weight (g) of the membrane, respectively.

#### 2.4.3. Water Uptake and Linear Expansion Ratio

Water uptake was measured to investigate the hydrophilicity of the membrane. Small pieces of the membranes were dried in a vacuum oven at 65 °C for 24 h. The dried pieces of the membranes were weighed and then soaked in deionized water at room temperature for 24 h. The wet membrane pieces were weighed after wiping out the extra water from the membrane surface with a blotting paper. The water uptake (%) was determined [42] using Equation (2):(2)$$\mathrm{water}\;\mathrm{uptake}\;\left(\%\right)=\frac{W_{\mathrm{wet}}-W_{\mathrm{dry}}}{W_{\mathrm{dry}}}\times 100$$
where W_wet_ and W_dry_ represent the weight of wet and dry membranes, respectively.

The linear expansion ratio (LER) was calculated to determine membrane stability. The dry membrane samples (3 cm × 1 cm) were immersed in deionized water for 24 h. LER (%) was measured using the following Equation (3) [43]:(3)$$\mathrm{LER}\;\left(\%\right)=\frac{L_{\mathrm{Wet}}-{\;L}_{\mathrm{dry}}}{L_{\mathrm{dry}}}\times 100$$
where L_dry_ and L_wet_ represent the length of dry and wet membranes, respectively.

#### 2.4.4. Thermal Stability

The thermal properties were characterized by thermo gravimetric analysis (TGA) using a TGA Mettler Toledo analyzer at 10 °C/min heating rate; the temperature program used for TGA involved a temperature increase from room temperature to 800 °C. Nitrogen gas with a flow rate of 15 mL/min was used to obtain an inert atmosphere.

#### 2.4.5. Mechanical Properties

Mechanical stability of QDAB hybrid membranes was determined with the help of the U-CAN Dynatex tensile tester (UT-2080, Taiwan) having crosshead speed of 100 mm/min. The dumbbell-shaped sample had a length of 25 mm between the jaws and a width of 5 mm. The experiment was performed at a relative humidity of 25 ± 2% and at a temperature of 25 °C.

#### 2.4.6. Scanning Electron Microscopy

The morphology of the prepared membranes was examines through SEM using Jeol (JSM-6490A). The surface images of the membranes were collected under low vacuum, at a voltage of 5 KV.

#### 2.4.7. Acid Recovery Experiment

The DD experiment was performed using a cell with two compartments, which were divided by the membrane, with an effective area of about 6.2 cm^2^. The membranes were dipped in the feed solution for 2 h before the test. One compartment of the dialysis cell was filled with 100 mL of feed solution consisting of a HCl/FeCl_2_ aqueous mixture (1 M HCl/0.25 M FeCl_2_), and the other compartment was filled with 100 mL of deionized water as the dialysate. Both compartments of the dialysis cell were stirred continuously with the help of magnetic stirrer at an equal speed to avoid concentration polarization. The experiment was performed at 25 °C for 1 h, and then from both compartments of the cell, feed and permeate solutions were removed. The concentration of H^+^ and Fe^2+^ ions in feed and permeate were measured by titration using standard solutions of Na_2_CO_3_, with methyl orange and KMnO_4_ as indicators, respectively. 

The dialysis coefficient (U) was calculated using the following Equation (4) [38]:(4)$$U=\frac{M}{At\Delta C}$$
where *M* indicates the amount of component transported in moles, *A* is the effective area of the membrane (m^2^), *t* stands for time in h, and ΔC is the logarithm average concentration (mol/m^3^) between the two compartments, which is given by Equation (5) [36,38]:(5)$$\Delta C=\frac{C_{f}^{0}-\left(C_{f}^{t}-C_{d}^{t}\right)}{\ln\left[C_{f}^{0}/\left(C_{f}^{t}-C_{d}^{t}\right)\right]}$$
where $C_{f}^{0}$ and $C_{f}^{t}$ are the concentrations of the feed solution at time 0 and t, respectively, whereas the concentration of the dialysate at t time is $C_{d}^{t}$.

The separation factor (S) was determined by the ratio of (two dialysis coefficients) the acid dialysis coefficient (U_H_) and the dialysis coefficient of ferrous ion (U_Fe_) in the solution. The S value was determined using Equation (6):(6)$$S=\frac{U_{H}}{U_{\mathrm{Fe}}}$$

## 3. Results and Discussion

### 3.1. FTIR Spectra Analysis

The FTIR spectra of pristine PVA, QDAB, TEOS, and cross-linked QDAB/PVA AEMs are shown in Figure 1. In the spectra of QDAB, the peaks at 1925 and 1645 cm^−1^ correspond to the stretching of –CH_2_ and –CH, and the peak at 1469 cm^−1^ is due to the stretching of C−N from the ammonium groups. These peaks are also observed in the spectra of the prepared membranes. The band in the 3100–3550 cm^−1^ region in QDAB and PVA corresponds to the stretching vibration of −OH groups [38]. In cross-linked QDAB/PVA AEMs, the absorption band in the region 1255–1350 cm^−1^ represents the secondary C−N stretching vibration of QDAB [44]. The strong and broad band in the 3100–3550 cm^−1^ region is assigned to the stretching vibration of −OH groups [39]. The band of the −CH stretching vibration of alkyl groups is observed around 2925 cm^−1^ [45]. The characteristics band of the quaternary ammonium [–N^+^(CH_3_)_3_] group is observed at 1469 cm^−1^ [13,38]. The stretching bands of Si−O−C and Si−O−Si linkages are observed in the region from 1052 to 1150 cm^−1^ [46,47], while a small peak at 945 cm^−1^ is attributed to the stretching vibration of the silanol (Si−OH) group [48]. The intensity of the stretching vibration of the –OH band increased with the increase in the QDAB content, as shown in Figure 1. The Si−O−C groups were produced from the sol–gel reaction between the hydrolyzed silanol (Si−OH) of TEOS. The Si−OH groups reacted with the −OH groups of QDAB or PVA through a cross-linking reaction, forming the Si−O−C group. Hence, the presence of Si−O−C and Si−O−Si linkages confirmed the condensation and cross-linking reaction between TEOS, PVA, and QDAB, which led to the development of a cross-linking network between the inorganic and organic components

### 3.2. Ion-Exchange Capacity

The IEC value was measured using Mohr’s titration method to confirm the charged nature of the AEMs. The influence of QDAB content on the IEC value is depicted in Figure 2 and reported in Table 1. The IEC represents the charged nature of the membrane and is directly proportional to functional groups fixed in the membrane matrix. The IEC was mainly contributed by the −N^+^(CH_3_)_3_Cl^−^ functional groups present inside the membrane matrix. The IEC values of the QDAB membranes increased from 0.86 to 1.46 mmol/g with the increase in the content of QDAB. These IEC values are comparable to or higher than those reported for different AEMs [23,32,49,50].

The IEC values of QDAB AEMs increased with the increase in the amount of QDAB, resulting in greater charge density. These results indicated the successful cross-linking of QDAB with the PVA chain; the uncross-linked QDAB was washed out during the membrane fabrication process and showed a constant IEC value.

### 3.3. Water Uptake and Linear Expansion Ratio

The water uptake and linear expansion ratio of the prepared AEMs are presented in Table 1 and Figure 3. The water uptake values of QDAB AEMs were in the range of 71.3–147.8%. Moreover, the results showed that the water uptake values increased by increasing the content of quaternary ammonium groups in hybrid PVA blend membranes. The water uptake of QDAB membranes showed an increasing trend similar to that of the IEC values, and QDAB-70 exhibited the highest water uptake values. The higher hydrophilic nature of the AEMs assisted the transport of ions [13], and more ions were transported inside membrane matrices with high water uptake.

The linear expansion ratio (LER) is used to estimate the dimensional stability of AEMs during the DD process. As shown in Figure 3, as the QDAB loading (30 wt. % to 70 wt. %) increased in the PVA matrix, LER increased from 28.7 to 39.5%. These results indicate the flexibility of the AEMs, which means that the membranes are not brittle and showed swelling resistance. TEOS reacted with −OH group of PVA via the sol–gel reaction, with the formation of Si−O−Si and Si−O−C linkages which can significantly improve membrane stability [15]. Hence, these bonds supplied strength to resist linear expansion. It was found that the membrane with the highest water uptake showed the higher LER. TEOS was used as a cross-linker, and the amount of TEOS was the same for all prepared membranes; the QDAB content increased, which produced a decrease in the degree of cross-linking in the membrane. Therefore, the cross-linking degree decreased in membranes with a higher QDAB amount and led to an increase in LER.

### 3.4. Thermal Stability of QDAB AEMs

The thermal stability of the QDAB members was evaluated using TGA, and the results are presented in Figure 4. Each thermogram shoes mainly four weight loss steps. The first step was associated with the removal of bound and unbound water from the membrane matrix, dequaternization occurred in the second step, the next step involved the removal of hydroxyl groups from PVA, and finally the degradation of the membrane matrix was completed. The first step of weight loss in the temperature range of 25–200 °C is attributed to dehydration, i.e., the removal of bounded or unbounded water molecules and water produced from the condensation reaction during the sol–gel process from the membrane matrix [28]. With the increase in the amount of QDAB in the PVA matrix, the hydrophilicity of the membrane increased; therefore, in this step, the QDAB-70 membrane showed the largest weight loss in comparison with the other QDAB membranes. The weight loss in the second step from 200 to 330 °C is associated with the decomposition of quaternary ammonium groups and the rupture of the PVA chain. Here, QDAB-70 exhibited a higher weight loss as compared to the other prepared AEMs. This was due to the increase in the amount of −N^+^(CH_3_)_3_ functional groups which started to degrade at a relatively low temperature [51]. The next step from 330 to 490 °C is attributed to the elimination of hydroxyl groups from PVA [52]. In this step, increasing the content of QDAB in the membrane matrix improved the thermal stability of AEMs. The final degradation stage from 490 °C onwards is the carbonization of the main backbone of polymer chains to ash residue [53]. The incorporation of QDAB functionality in PVA enhanced the membranes’ thermal stability in this step.

The TGA thermograms also demonstrated that QDAB has a vital role in thermal stability, and the prepared QDAB AEMs are thermally stable in nature. Initially, the QDAB-70 membrane demonstrated higher weight loss compared to the other membranes. as shown in Table 2. Afterward, the QDAB-70 membrane became thermally stable and showed less weight loss compared to QDAB-60 in the degradation stage. Similarly, QDAB-50 and QDAB-40 AEMs underwent a more limited weight loss compared to the QDAB-30 membrane. For example, a 40% weight loss was observed at 381, 376, 371, 365, and 358 °C for the QDAB-70, QDAB-60, QDAB-50, QDAB-40, and QDAB-30 membranes, respectively. Similarly, 60% weight loss was observed at 441, 438, 421, 411, and 396 °C for the QDAB-70, QDAB-60, QDAB-50, QDAB-40, and QDAB-30 membranes, respectively.

### 3.5. Mechanical Strength of QDAB Membranes

The influence of QDAB content on the mechanical properties of the membranes is shown in Figure 5. The TS values of the prepared AEMs were in the range of 26.1–41.7 MPa, whereas the Eb values varied from 68.2% to 204.6%. These values are much higher than those of hybrid PVA/SiO_2_ anion exchange membranes with multi-silicon copolymer (TS: 7–12 MPa, Eb: 42–97%) [49], which indicates the superior mechanical strength of our QDAB hybrid membranes compared to other membrane described in the literature.

It has been found that an increase in the content of QDAB resulted in a decrease of TS and an improvement of Eb values. The QDAB-70 membrane had high flexibility and showed a maximum Eb value as well as a low TS value. The enhancement of the Eb values with the maximum dosage of QDAB is attributed to soft hydrophilic sites in the QDAB compound, which acts as a diluent that reduces the interaction between the long chains of PVA. Thus, it reduces the tensile strength and enhances the mobility of the PVA chains, leading to the maximum Eb value of the QDAB-70 membrane [54]. Similar results have been reported for hybrid membranes and attributed to the miscibility and cross-linking of organic and inorganic moieties inside a hybrid polymeric membrane [55]. 

### 3.6. Morphological Analysis

The influence of QDAB content on membrane morphology was studied by SEM and arise presented in Figure 6. SEM images suggested that the membrane surface was compact and dense, without any holes or cracks. Figure 6e shows some aggregates on the surface of the QDAB-70 membrane due to the accelerated hydrolysis of TEOS with the increase in QDAB content. It was observed that QDAB was uniformly distributed promoting a homogeneous structure without any evidence of phase separation. The homogeneity of the AEMs indicated the compatibility between the QDAB anion exchange precursor and the PVA matrix. The QDAB anion precursor is effectively cross-linked between PVA and TEOS with covalent and hydrogen bonds through the sol–gel reaction [34,36]. The aggregations of particles increased in the AEMs with the increase in the amount of QDAB, as compared other QDAB AEMs. In the QDAB-70 membrane, a slight phase separation was observed because of the high anionic content [32]. Therefore, a larger aggregation of silica-rich particles may occur in the QDAB-70 membrane.

### 3.7. Acid Recovery Performance of QDAB AEMs via Diffusion Dialysis (DD)

Acidic waste solutions mostly contain metal ions and inorganic acid, which is usually generated during different industrial processes such as metal electrolysis and metallurgical and acid cleaning processes [10]. Therefore, to analyze the DD performance properties for acid recovery, QDAB AEMs were employed to separate a HCl/FeCl_2_ aqueous solution (1 M HCl/0.25 M FeCl_2_) used as a model aqueous waste feed to assess their potential for application in acid recovery. The DD results of AEMs with different wt. % of QDAB in the PVA matrix in terms of acid dialysis coefficients (U_H_), separation factor (S,) and dialysis coefficient ferrous ion (U_Fe_) at 25 °C are shown in Figure 7.

As shown in Figure 7, the acid dialysis coefficients (U_H_) values of QDAB AEMs increased rapidly by increasing the content of QDAB, which was attributed to −N^+^(CH_3_)_3_ functional groups that enhanced the interaction between the membrane matrix and ions. The obtained U_H_ values were in the range of 0.0186–0.0295 m/h and were relatively higher than those of PVA/QUDAP AEMs (0.008–0.0222 m/h at 25 °C) [27]. It was found that the QDAB-70 membrane with 70 wt. % of QDAB content exhibited the highest U_H_ value of 0.0295 m/h. The U_H_ value of QDAB-70 AEM was higher than the U_H_ values of a multi-silicon/BPPO-based membrane (0.025 m/h) [22], a PVA-based hybrid AEM (0.018 m/h) [38], and a commercial DF-120 membrane (0.009 m/h) [34].

The mechanism of ion transport from the membrane matrix can be depicted by considering the action of diffusion dialysis, the hydrophilicity, hydrogen bonding, ion exchange of functional groups, and structure of the membrane matrix [36,49]. The Si−OH group from TEOS and −OH groups of PVA transport the H^+^ ions from the membrane matrix through hydrogen bonds. The existence of hydrogen bonds is beneficial and accelerate the migration of H^+^ ions through the membrane [36]. QDAB hybrid membranes possess −N^+^(CH_3_)_3_ functional groups that act as ion exchange sites to facilitate the migration of Cl^−^ ions. Therefore, an increase in the content of −N^+^(CH_3_)_3_ group in the polymeric PVA matrix accelerates the transport of ions. The −N^+^(CH_3_)_3_ groups are beneficial for anion transport, allowing Cl^−^ ion diffusion in the membrane. Therefore, the available H^+^ and Fe^2+^ cations also cross the membrane from feed side to dialysate side in order to meet the electrically neutral requirements. H^+^ ions are transported much more extensively compared to Fe^2+^ ion due to their small size and lower valence state [56,57]. 

It has been observed that the U_Fe_ values increased from the membrane QDAB-30 to the QDAB-70. This was due to the higher water uptake and IEC value of the QDAB-70 membrane. A high water uptake of QDAB membranes is also beneficial for the free movement of ions through the membrane matrix that results in the transport of more Fe^2+^ ions. A higher IEC facilitates the transport of counter anions through the ion exchange sites [58]. Hence, the increasing trend of U_H_ and U_Fe_ from QDAB-30 to QDAB-70 agrees well with the increase of IEC and of the water uptake values.

The separation factor is the ratio between U_H_ and U_Fe_, which indicates the difference in transport rates between H^+^ and Fe^2+^ ions. The obtained separation factor values were in the range of 24.7–41.4 and were greater than the reported value of hybrid PVA–SiO_2_ membranes (15.9–21) at 25 °C [32] and of commercial DF-120 AEMs having an S value of 18.5 [34]. The higher separation factor values are credited to the amplified selective interaction between the H^+^ ions and AEMs.

The S values decreased gradually from the membrane QDAB-30 to the QDAB-70. This decrease was mainly due to the increase of IEC and water uptake in the presence of a higher QDAB content, resulting in the freel passing of H^+^ and Fe^2+^ ions through the membrane without much resistance by Cl^−^. This unrestricted transport of Fe^2+^ ions is not favorable for membrane selectivity [7]. There is usually a trade-off between U_H_ and selectivity. For example, QDAB-30 AEM showed the highest S value of 41.4, while it exhibited the lowest U_H_ of 0.0186 m/h, while QDAB-70 showed the lowest selectivity (S = 24.7) and the highest U_H_ value (0.0295 m/h).

In Table 3, we compared our results about acid recovery with those reported in the literature for AEMs. From the comparison, it can be easily seen that the prepared QDAB AEMs perform better than or comparably to many other reported membranes and have potential for acid recovery applications in the industry.

## 4. Conclusions

In this work, novel hybrid AEMs were successfully synthesized with PVA, TEOS, and varying quantities of QDAB, from 30 to 70 wt. %, by the sol–gel reaction. The FTIR spectra confirmed the successful cross-linking of QDAB with PVA. IEC increased from 0.86 to 1.46 mmol/g, water uptake increased from 71.3–147.8%, and linear expansion ratio augmented from 28.7 to 39.5% with the increase of QDAB content in the membrane matrix. TGA analysis indicated that the membranes have good thermal stability. The tensile strength was 26.1–41.7 MPa, and flexibility values varied from 68.2% to 204.6%. SEM analysis showed that the membrane surface was compact and dense. The U_OH_ values increased from 0.0186 to 0.0295 m/h, while the S values decreased from 41.4 to 24.7 with the increase of the content of QDAB. QDAB/PVA hybrid AEMs showed high proton permeability and selectivity compared to the commercial DF-120 AEM. The prepared AEMs have great potential for application in DD for acid recovery from industrial wastewater.

## Data Availability

Not applicable.

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
