# Peer review of "Quaternized Diaminobutane/Poly(vinyl alcohol) Cross-Linked Membranes for Acid Recovery via Diffusion Dialysis"

_membranes, 2021, doi:10.3390/membranes11100786_

Round 1

Reviewer 1 Report

Overall, the manuscript is well written and describes the studied membranes and their characterization.

  • General editing should be improved.
    1. Line 30 “ MPa, and elongation”
    2. Line 38, “Acid wastewaters are…”
    3. Line 78 “et.al”
    4. etc...
  • What does “cleaner” mean in the context of line 48
  • Line 76-77 notes “In many previous…” but only cites one source. Additional should be provided.
  • Was any confirmation of complete conversion or residual precursors done?
  • Section 2.4.4 describes the test but not the material prep. Looking ahead at Figure 4 and Table 2, it is noted that removal of water is seen in the TGA data. How were the films treated prior to TGA? Fully hydrated? Dried in air? Dried in vacuum? Heated? Please provide.
  • Section 2.4.3: why use just the length change and not the full volumetric swelling?
  • Section 2.4.6 should be edited and improved to include precise details.
  • In Figure 2 and Table 1, IEC is reported. How does the measured IEC compare to the theoretical IEC based on the membrane chemistries?
  • The SEM images of the surface are quite blurry and not sufficiently high quality, in my opinion, for publication. Also, why was the crosssection also not imaged to look at the through-plane morphology?
    • The text also notes how these are homogeneous, but there appear to be different surface features (particulates? and roughness that is not described or explained.
  • Why are Fig 7a and 7b different sizes?

Author Response

File has been attached

Reviewer 2 Report

The article entitled ‘Quaternized diaminobutane/Poly (vinyl alcohol) cross-linked membranes for acid recovery via diffusion dialysis’ reports the synthesis of anion exchange membrane using various wt% percent of quaternized diaminobutane with the PVA and TEOS by sol-el method. The present manuscript authors systematically tested the acid recovery study and explained their membrane efficiency. However, it seems experimental data and explanation missing in the present form of the manuscript. Therefore, I recommend this manuscript to be considered for publication after having the following minor points clarified.

  1. In page 6, section 3.1 FTIR spectra, compare pristine PVA, QDAB, and TEOS with as prepared membrane band changes.
  2. What is the observation of Fig. 1, upon QDAB wt % increase?
  3. Page 12. Fig. 6 SEM image scale bars are not visible.
  4. Table 3 reference 22 seems like better performance with present work, justify this.

Author Response

File has been attached
